# Assessment of University Students on Online Remote Learning during COVID-19 Pandemic in Korea: An Empirical Study

**Ji-Hee Jung [1] and Jae-Ik Shin [2,\*]**

[1]  Department of Business Administration, University of Ulsan, 93 Daehak-ro, Nam-gu, Ulsan 44610, Korea; aboutjee@ulsan.ac.kr
[2]  Department of Smart Distribution and Logistics, Gyeongsang National University, 33 Dongjin-ro, Jinju 52725, Gyeongnam, Korea
\*  Correspondence: sji15@gnu.ac.kr; Tel.: +82-55-772-3663

**Abstract:** The COVID-19 pandemic is affecting business, society, and education worldwide. In particular, it is an example of a non-face-to-face approach in commerce and education. As the pandemic has lasted for nearly two years in Korea, online remote learning has been held at universities for four semesters. There are learning management systems (LMSs) and real-time Zoom lectures for the types of online remote learning widely used at Korean universities, and students and professors are adapting with some difficulties due to the unfamiliar lecture environment. Assessment of students' online remote learning is essential in reinforcing strengths and compensating for weaknesses. This study aimed to investigate the relationship between online remote learning quality (system quality, information quality, and service quality), flow, and learner satisfaction, and the implications are presented. A survey was conducted on 182 university students, and the structural equation model of AMOS 21.0 was used to analyze the research hypothesis. The results of the empirical analysis were as follows. First, system quality, information quality, and service quality had a positive effect on flow. Second, system quality, information quality, and service quality had a positive effect on learner satisfaction. Third, flow had a positive effect on learner satisfaction. It seems that online remote learning in university is becoming a new normal. In conclusion, the implications and limitations of this study are presented.

**Keywords:** COVID-19 pandemic; online remote learning quality; flow; learner satisfaction

## 1. Introduction

As the inflow and spread of the COVID-19 pandemic in Korea became rapid, in February, which was in a severe stage, the government (Ministry of Education) urgently announced measures to postpone school opening. Since then, face-to-face classes at elementary, middle, and high schools nationwide were completely banned for three months unprecedentedly [1,2]. Such a situation is a common phenomenon in Korea and the world education field [3,4].

Information technology has begun to gain momentum as students have faced difficulties in learning due to the closure of educational institutions by the pandemic. Even during quarantine, information technology provides a solution for the continuous learning process through innovative LMSs [5,6].

LMSs are provided through a cloud-based service and are used in non-face-to-face classes to give a variety of classes reflecting the needs of learners. The LMS is similar to e-learning in that it uses the internet and programs to learn. The system convenience of the LMS increases users' access to learning, and the system quality enhances user participation and thus affects academic achievement [7–10].

It allows professors to evaluate students' coursework and implement IT solutions for teaching. Faculty, students, and education authorities are making appropriate use of the technology and efficient learning process [11]. With the COVID-19 pandemic, existing

offline classes have been replaced by online courses, and online learning at educational institutions had an opportunity to rise rapidly [3]. With the development of technology, online classes have been able to respond effectively to the needs of various learners, such as online interaction between instructors and learners and customized learning [12].

When it comes to online classes, Korea has a relatively low barrier to entry compared to other countries in the world. Based on its excellent infrastructure and networks, the government has continuously supported e-learning policies and related industry development for a long time [13]. In the school field, e-learning, blended and flip learning, multimedia, and intelligent and ubiquitous education have been emphasized to improve practical teaching and learning ranging from teacher training to university evaluation indicators [14–16].

Significant issues such as non-face-to-face classes, remote online education, e-learning, and online classes, which are emerging as core agendas in the transformation of the education system due to the COVID-19 pandemic outbreak, have previously been studied and practiced mainly in the field of pedagogy. In recent years, it has become a social phenomenon that all members of society pay attention to and experience [17].

In 2020, Korean universities were offering online classes instead of face-to-face classes due to the COVID-19 pandemic. Online remote learning means that professors and students are not in the same place and are taught the same content as existing face-to-face courses using the internet for broadcasting. Although we have to watch the trend of the COVID-19 pandemic, university remote learning was still being held in most lectures in the second semester of 2020. Several universities have announced that they will alternate between the remote and face-to-face classes in the second semester or take face-to-face learning courses that require experimentation and practical skills. Most of the other universities internally conduct academic management similar to the first semester.

Previously, in Korea, the LMS was not used in all classes, but due to the COVID-19 situation, most classes now have to be conducted non-face-to-face, and different techniques and methods for conducting online assessments should be explored [18]. There were difficulties in the early stages of implementation due to a lack of experience and insufficient establishment of class content and infrastructure. However, it has been evaluated that the university has established itself to a certain extent thanks to the swift response of the university and the active participation of students [19]. Ribble [20] said that, in reality, not all students are technically savvy, and not all instructors are incompetent. In addition, apart from technical capabilities, conditions directly related to the satisfaction of online lectures, such as digital devices, learning situations, and comfort of the place, were found to be different for each learner [20]. Nevertheless, the LMS is evolving toward supporting interactive and learning experiences optimized for various online learners and appropriate to the student's level [21].

According to a survey conducted by 6261 university students of 27 Korean National University Student Association Networks from 18 to 30 April 2020, the percentage of students who answered that they were satisfied with the online classes was found to be 6.8%. Research on the effectiveness of remote online education has been conducted in various ways over a long period. The results of accumulated research on remote education using ICT have played a large role in developing learning tools. However, most of the students are not satisfied with the quality of the remote lectures to deal with the COVID-19 pandemic. It is difficult to effectively conduct online remote lectures because the foundation for real-time has not been sufficiently established.

With the prolonged COVID-19 pandemic, online remote learning is expected to continue in the second semester of 2021. The evaluation of online remote learning of university students through the LMS and Zoom for the last two years showed that they were satisfied with the class and showed a sense of social isolation and limitations in face-to-face interaction. Nevertheless, alternatives that can increase flow and learner satisfaction in the routine online remote learning will be needed. It would be wise to evaluate the quality of online remote learning based on the LMS by university students and find improved solutions.

As remote learning can be a new normal, exploring constructs affecting online remote learning quality and learner satisfaction is essential. In this study, based on previous research, the relationship between the remote class quality (system quality, information quality, and service quality), flow, and learner satisfaction was investigated. Through this, the theoretical and practical implications for the effective operation of the remote lectures were presented.

## 2. Theoretical Background and Hypothesis

### 2.1. Online Remote Learning Quality

Jang [22] classified the convergence of education and ICT (Information and Communication Technology) as ICT using education, e-learning, m-learning, u-learning, and smart learning according to the learning type, major services, and major devices. Among them, e-learning implements cyber home learning through the LMS based on internet PCs, or mobile phones are used as the main technology. Therefore, the online learning currently conducted in Korea can be mainly viewed in the form of e-learning. Recently, in line with the easing of social distancing, education is being conducted in the form of blended learning, in which school attendance and online learning are combined to respond to COVID-19 and, at the same time, to facilitate education [23].

Online remote classes can be classified into real-time classes and non-real-time classes depending on the time the lectures are held. The online courses were held non-real-time for several weeks after classes in the first semester of 2020. After the first semester, full online classes were decided, real-time lectures were conducted, non-real-time lectures were maintained, or both methods were used together [24].

According to Hodges et al. [25], online classes in crises such as the COVID-19 pandemic are called emergency remote teaching (ERT). They pointed out that, fundamentally, it should be viewed differently from existing online learning. As it is an emergency, it requires a flexible approach throughout education, so it refers to asynchronous rather than synchronous, and the flexibility of operation such as task deadlines, subject policies, and school policies.

Online remote lectures at domestic universities have also been continuously expanding. In order to keep up with the latest trends in university education, large-scale open online lectures such as MOOCs (Massive Open Online Courses) have been developed, and the limitations of the educational environment such as securing lecture spaces have been overcome. For this purpose, online remote lectures are increasing [26]. However, distance lectures at domestic universities have been mainly used for large-scale lectures according to the regulation of education policy [27], and the proportion of total lectures is also very small, so it is not possible to effectively introduce and establish distance lectures in various types of classes.

In this situation, the sudden conversion of all lectures to online remote lectures due to the epidemic of an infectious disease such as COVID-19 is causing many difficulties for all university members. Despite the negative learning experience of university students compared to face-to-face classes, it is expected that non-face-to-face classes will be more active with COVID-19 and post-COVID-19 eras. Therefore, online remote classes are now becoming a necessity, not an option [28].

The rapid development of the internet has made online learning possible for millions of children and adults worldwide [29]. Learning contents and methods are also becoming more diverse than ever before [30]. This enables learners and instructors to learn remotely and provide education anywhere without physical constraints [31]. Although COVID-19 has forced students to take online remote learning, they reap benefits such as transportation cost reduction and better time use. It can be estimated that the non-face-to-face classes have brought about a significant carbon reduction effect as university students across the country do not need to use a vehicle to visit school.

Delone and McLean [32] presented an information system success model (ISSM) and analyzed the relationship between system quality, information quality, usage level, user

satisfaction, personal performance, and organizational performance. After that, service quality was added to the system and information quality in consideration of the changing environment [33].

Today's students are seen as customers of universities, and they should provide the best e-learning service quality to their students [34]. Previous studies in the field of e-service quality have provided a logical point of departure for future research. Lin [35] used DeLone and McLean's [32] information system success model to find the factors that lead to the success of e-learning systems in Taiwan and found that three factors, namely, system quality, information quality, and service quality, have impacts on the success of the systems.

Samarasinghe and Tretiakov [36] classified service quality, content quality, and system quality as e-learning quality. The competitiveness of e-learning can be achieved by combining the design, technical capabilities, excellent educational content, and continuous management for learners. In this study, the quality of the non-face-to-face class was classified into system quality, information quality, and service quality based on previous studies, and the measurement items for the quality concept were set.

(1) System Quality. The system refers to a system used by online remote lecture providers to manage and control a learning environment of e-learning services. The system quality of the remote lecture refers to the quality of hardware, including computers, internet networks, and security systems for students. Even if the learning contents are similar, the learners' satisfaction can differ depending on the technical method by which the remote lectures are delivered and operated. System quality becomes a key aspect that can be used to measure the success of an information system. Providers who provide and maintain services play an essential role in user satisfaction. Assessment based on user feedback will improve the quality of e-learning. This action will benefit the institution in the long run [37].

(2) Information Quality. As online remote lectures are conducted based on education content, information quality is a fundamental element of lecture success and affects learner satisfaction. Managing information quality should help instructors and learners achieve the instructional design and learning goals using suitable multimedia materials and interfaces. Education contents that consider students' characteristics and learning time, instructional and learning designs that induce their autonomous learning, diverse and meaningful interactions, and evaluation considering the learning situation and degree are necessary [38]. The clearer and more valid the educational content, the more positive the learning effect is, so the learner feels that e-learning is effective and satisfied [39]. In addition, the good information quality of the e-learning system can increase user satisfaction and the utilization of the system by students [40].

(3) Service Quality. Service quality can be said to represent the service capability of online remote lectures. It refers to the overall support and operation management provided by the service provider. To manage the quality of the remote lecture service, professional staffing allocation, infrastructure for stable service provision, educational and administrative support through learner satisfaction surveys, and a consortium of inter-university and industry-academia for co-development of the courses are necessary. Service quality is closely related to customer satisfaction, as it is the driving force behind a strong relationship with the company [41]. Previous studies have identified e-service quality as a significant factor in increasing customer satisfaction [42]. It is assumed that the service provided is good and can generate satisfaction, in which case the customer indirectly decides to use the product or service for a more extended period, commonly referred to as stickiness [43].

### 2.2. Flow

Flow refers to the optimal psychological phenomenon that occurs when you focus on activity [44]. It is the experience of solid concentration, integration of behavior and awareness, loss of self-awareness, personal control, distortion of time, and intrinsic reward.

It can be called the self-purpose experience. The concept of flow can be observed a lot in sports, entertainment, shopping, and computer games. It has been suggested as a concept to explain consumption behavior in the internet environment [45].

In a computer-mediated environment, flow experience is reported as a continuous and enjoyable loss of self-consciousness and self-reinforcing behavior through interaction with machines. During flow experience, consumers become a state of behavior transfer, behavior control, and self-engagement, which can be a key concept to understand consumer behavior and experience in an online environment [46]. Flow in learning is a psychological mechanism that promotes concentration and participation, leading to high academic achievement [45]. It was found that the more frequently flow is experienced in the learning process of e-learning, the higher the learning performance is [47].

Learning flow is a useful variable that can directly or indirectly predict the learning process and results in a state where the learner is completely focused on learning [48,49]. Online classes are in an environment where there is no contact with the instructor, and there is the possibility that learners will not be able to concentrate on the class. Learner satisfaction can be enhanced when learning flow is increased by presenting clear learning goals, providing them with a voluntary atmosphere and opportunities for interaction [50].

*2.3. Learner Satisfaction*

Learner satisfaction is the positive level of the learner's educational experience, and it is the learner's perception of the success and results achieved. In addition, learning satisfaction can be said to be a degree of positive recognition of the relationship between the remote lectures and the learning experience [51]. Satisfaction is an indicator that reveals that the learner's needs have been met. If you participate in the learning process to create knowledge and feel its value and use the result, it can be said that your satisfaction is improved [52].

Perceived quality in e-business is a significant influencing factor on customer satisfaction, and the quality of online remote lectures is also an essential factor in learner satisfaction. Learner satisfaction is the response of learners who participated in the remote lectures, affecting their attitudes toward e-learning. In other words, it is a significant factor influencing e-learning evaluation and leads to learner loyalty (continuous use and word of mouth). This learner satisfaction has been used as an essential factor for evaluating educational quality [53,54].

It has been reported through several studies that flow in a learning environment increases the learning effect. Flow means that learners can recognize learning as a pleasant experience and achieve high academic achievement. Satisfaction results from the pleasure of attention, relevance, and confidence and is the most common index for measuring learning outcomes and can be related to the persistence of learning motivation [52,55].

In this study, flow and learner satisfaction were used as indicators to measure the performance of online remote learning in the COVID-19 pandemic situation, and online remote learning quality was used as a factor influencing this performance. Therefore, based on previous studies, the following hypotheses were established considering the influence relationship between the quality of online remote learning (system quality, information quality, and service quality), flow, and learner satisfaction.

**Hypothesis 1 (H1).** *System quality will have a positive effect on flow.*

**Hypothesis 2 (H2).** *Information quality will have a positive effect on flow.*

**Hypothesis 3 (H3).** *Service quality will have a positive effect on flow.*

**Hypothesis 4 (H4).** *System quality will have a positive effect on learner satisfaction.*

**Hypothesis 5 (H5).** *Information quality will have a positive effect on learner satisfaction.*

**Hypothesis 6 (H6).** *Service quality will have a positive effect on learner satisfaction.*

**Hypothesis 7 (H7).** *Flow will have a positive effect on learner satisfaction.*

### 3. Research Method

During the two years of the COVID-19 pandemic in Korea, many universities conducted non-face-to-face classes using the LMS. University students can access the LMS using a PC, laptop, smartphone, or mobile device to take online remote classes anytime, anywhere. Although there are many opinions about the performance between non-face-to-face classes and face-to-face classes, in reality, many efforts are underway to improve the performance of non-face-to-face classes, which have been chosen as the next best thing due to the COVID-19 pandemic. Thus, this study examined the relationship between online remote lecture quality (system quality, information quality, and service quality), flow, and learner satisfaction based on previous studies. It was conducted on students who participated in non-face-to-face classes due to the COVID-19 pandemic, and the research model is shown in Figure 1.

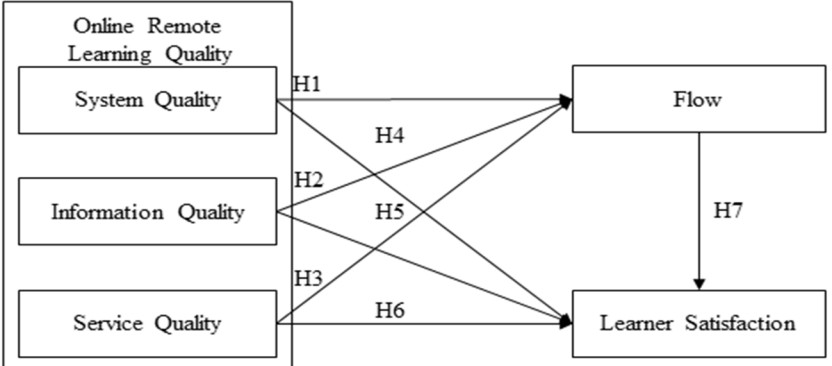

**Figure 1.** The research model.

A self-administered questionnaire was developed for this study. The survey included the perception of system quality, information quality, and service quality of online remote learning, flow, and learner satisfaction. The questionnaire consisted of four parts: (1) online remote learning (system quality, information quality, and service quality); (2) flow; (3) learner satisfaction; and (4) demographic information. The questionnaire items in this study are shown in Table 1. For the development and validation of the measurement instrument, the questionnaire items were borrowed from previous studies [2,3,12]. The 7-point Likert scale was used to measure online remote learning (system quality, information quality, and service quality), flow, and learner satisfaction. A nominal scale was used for the demographic characteristics scale. The measurement scale for each construct was modified and supplemented by applying it to this study using the previously tested items for validity and reliability. The survey participants were university students from Ulsan in South Korea who participated in online remote learning of the LMS.

### 4. Results

*4.1. Sample Characteristics*

The survey in this study was conducted for students taking online remote lectures at the department of business administration, U University in the first semester of 2020 from June to September 2020. The survey method was conducted through the internet survey of Google Forms. U university is a comprehensive private university located in Ulsan, South Korea, and has about 15,000 students. U university supports recorded lectures and real-time lectures using the open-source learning management system, Moodle, and Zoom video conferencing platform. Through the LMS, lectures can be viewed on the Internet and

mobile, and due to the nature of the recorded lectures, repetition learning is possible at the desired time of every week.

Of the 205 collected questionnaires, 182 were finally used for the empirical analysis. SPSS 25.0 and analysis of moment structure (AMOS 21.0) were used to analyze the collected data. There is some discrepancy regarding the recommended sample size for structural equation models. Bagozzi and Yi [56] would have to say that rarely (e.g., in a factor analysis of a small number of items with "well-behaved data") would a sample size below about 100 be meaningful and that one should endeavor to achieve a sample size above 100, preferably above 200. When there are three or more indicators per factor, a sample size of 100 is usually sufficient for convergence, and a sample size of 150 is usually sufficient for convergence and a suitable solution [57]. Therefore, the sample size of 182 used in this study can be appropriate for analysis.

The characteristics of the sample of 182 respondents used in the final analysis were as follows. The respondents were not incumbents, but full-time students. Among the demographic characteristics, the gender was 77 males (42.3 percent) and 105 females (57.7 percent). The second grade was 112 (61.6 percent), the third was 21 (11.5 percent), and the fourth was 49 (26.9 percent). As for the remote lecture method, 158 (86.8 percent) students preferred recorded lectures, and 24 (13.2 percent) preferred real-time classes.

### 4.2. Reliability and Validity Analysis

Consider the problem of how to measure a theoretical construct in a study. A confirmatory factor analysis (CFA) model is helpful in this regard. A more rigorous approach is to conduct a CFA. It formally tests whether a set of indicators share enough common variance to be considered measures of a single factor [56]. CFA was performed to test the measurement model using AMOS 21.0. It played an important role in securing strict convergent validity and unidimensionality among all the research constructs.

The final set of 15 items showed construct reliabilities, average variances extracted (AVEs) [58], and Cronbach's alphas that exceeded recommended standards for reliability and unidimensionality (see Table 1). All composite reliabilities exceeded the criterion value of 0.7, indicating that the measurement model has good reliability of the research constructs [59]. The AVE was also found to exceed the criterion value of 0.5, indicating that the model has convergent validity [60].

**Table 1.** Confirmatory factor analysis.

| Construct | Items | Standardized Loadings | S.E. | t-Value | Composite Reliability | Cronbach's $\alpha$ | AVE |
|---|---|---|---|---|---|---|---|
| System Quality | SY1. The remote lecture system has clear and reliable quality characteristics. | 0.763 | - | - | 0.876 | 0.856 | 0.702 |
| | SY2. The remote lecture system always responds quickly. | 0.816 | 0.099 | 10.95 | | | |
| | SY3. The remote lecture system delivers information in a clear and concise manner to both men and women of all ages. | 0.883 | 0.104 | 11.302 | | | |
| Information Quality | I1. Remote lecture providers provide information related to my learning. | 0.843 | - | - | 0.872 | 0.857 | 0.697 |
| | I2. Remote learning providers provide very easy-to-understand educational content. | 0.904 | 0.077 | 13.754 | | | |
| | I3. The remote lecture provider provides the latest information for my purpose. | 0.733 | 0.084 | 11.127 | | | |
| Service Quality | SE1. The instructor has enough knowledge to meet the needs of the learner. | 0.794 | - | - | 0.844 | 0.836 | 0.645 |
| | SE2. Instructors and operators respond kindly to the needs of learners. | 0.738 | 0.098 | 9.729 | | | |
| | SE3. Instructors and operators give individual attention to learners. | 0.844 | 0.096 | 10.477 | | | |



**Table 1.** *Cont.*

| Construct | Items | Standardized Loadings | S.E. | t-Value | Composite Reliability | Cronbach's α | AVE |
|---|---|---|---|---|---|---|---|
| Flow | F1. Time passed very quickly during the remote lecture. | 0.704 | - | - | 0.878 | 0.85 | 0.707 |
| | F2. While taking remote lectures, I am not affected by the surrounding conditions. | 0.906 | 0.124 | 11.24 | | | |
| | F3. Remote lectures are more focused than offline learning. | 0.829 | 0.115 | 11.033 | | | |
| Learner Satisfaction | LS1. I am satisfied with the overall remote lectures. | 0.835 | - | - | 0.906 | 0.885 | 0.764 |
| | LS2. I am satisfied with the recently learned remote lecture. | 0.919 | 0.07 | 15.085 | | | |
| | LS3. I think taking remote lectures was a good thing. | 0.797 | 0.072 | 13.173 | | | |

$\chi^2 = 119.057$, df = 75, $p = 0.000$, $\chi^2/\text{df} = 1.59$, GFI = 0.923, AGFI = 0.877, RMR = 0.046, TLI = 0.961, CFI = 0.972, RMSEA = 0.057.

The results are shown in Table 1. The overall model fit indices of the measurement model were $\chi^2 = 119.057$, df = 75, $p = 0.000$, $\chi^2/\text{df} = 1.59$, GFI = 0.923, AGFI = 0.877, RMR = 0.046, TLI = 0.961, CFI = 0.972, and RMSEA = 0.057, indicating that the model is generally suitable. With some consensus in the psychometric literature, it has been suggested that the model exhibits a good fit if the statistic adjusted by the degrees of freedom does not exceed 3.0 ($\chi^2/\text{df} \leq 3$) [61].

*4.3. Correlation Analysis*

The CFA confirmed the unidimensionality of the constructs. In order to test the discriminant validity, direction, and related degree between the constructs, correlation analysis was conducted. The results are shown in Table 2. As the values of the square root of AVEs were more than 0.7 and appeared higher than the values of the correlation coefficients, the criterion validity and discriminant validity of the measurement tool of this study were satisfied at the same time [58].

**Table 2.** Discriminant validity.

| Construct | (1) | (2) | (3) | (4) | (5) |
|---|---|---|---|---|---|
| (1) System Quality | **0.838** | | | | |
| (2) Information Quality | 0.301 ** | **0.835** | | | |
| (3) Service Quality | 0.240 ** | 0.203 ** | **0.803** | | |
| (4) Flow | 0.380 ** | 0.338 ** | 0.260 ** | **0.841** | |
| (5) Learner Satisfaction | 0.423 ** | 0.489 ** | 0.359 ** | 0.527 ** | **0.874** |

The diagonal bold is the square root value of the AVEs.

*4.4. Hypothesis Testing*

Structural equation modeling (SEM) allowed us to investigate the causal relationships between constructs in the model and to test the model against the obtained measurement data to identify how well the proposed model fit the data [62]. SEM is an appropriate statistical method for examining the hypothetical relationship between constructs proposed in this study. The SEM of AMOS 21.0 was used to test the hypotheses. The overall fit indices of the path model were shown as follows: $\chi^2 = 126.932$, df = 76, $p = 0.000$, $\chi^2/\text{df} = 1.670$, GFI = 0.919, AGFI = 0.872, RMR = 0.046, TLI = 0.955, CFI = 0.967, and RMSEA = 0.060,

indicating that the model is suitable as they meet the criteria. The results of the path analysis of this study are shown in Table 3.

**Table 3.** The results of hypothesis testing.

| | Hypothesized Path | | Std. Estimate | S.E. | t-Value | *p*-Value | Results |
|---|---|---|---|---|---|---|---|
| H1 | System Quality → Flow | | 0.237 | 0.078 | 3.024 | 0.002 | Accepted |
| H2 | Information Quality → Flow | | 0.248 | 0.087 | 2.860 | 0.004 | Accepted |
| H3 | Service Quality → Flow | | 0.178 | 0.082 | 2.167 | 0.030 | Accepted |
| H4 | System Quality → Learner Satisfaction | | 0.161 | 0.061 | 2.664 | 0.008 | Accepted |
| H5 | Information Quality → Learner Satisfaction | | 0.300 | 0.071 | 4.213 | 0.000 | Accepted |
| H6 | Service Quality → Learner Satisfaction | | 0.198 | 0.065 | 3.026 | 0.002 | Accepted |
| H7 | Flow → Learner Satisfaction | | 0.284 | 0.068 | 4.155 | 0.000 | Accepted |

$\chi^2$ = 126.932, df = 76, *p* = 0.000, $\chi^2$/df = 1.670, GFI = 0.919, AGFI = 0.872, RMR = 0.046, TLI = 0.955, CFI = 0.967, RMSEA = 0.060.

The hypothesis H1 that system quality will positively affect flow was adopted as a standardization factor of 0.237 and t = 3.024. It shows that the higher the flow, the better the system quality. The hypothesis H2 that information quality will positively affect flow was adopted as a standardization factor of 0.248 and t = 2.860. It shows that the higher the flow, the more improved the information quality. The hypothesis H3 that service quality will positively affect flow was adopted as a standardization factor of 0.178 and t = 2.167. It shows that the higher the flow, the more improved the service quality. Thus, it can be seen that the three qualities of online remote learning quality are important strategic means to increase flow.

The hypothesis H4 that system quality will positively affect learner satisfaction was adopted as a standardization factor of 0.161 and t = 2.664. It shows that system quality is a major factor that can enhance learner satisfaction. The hypothesis H5 that information quality will positively affect learner satisfaction was adopted as a standardization factor of 0.300 and t = 4.213. It shows that information quality is a major factor that can enhance learner satisfaction. The hypothesis H6 that service quality will positively affect learner satisfaction was adopted as a standardization factor of 0.198 and t = 3.026. It shows that service quality is a major factor that can enhance learner satisfaction. The hypothesis H7 that flow will positively affect learner satisfaction was adopted as a standardization factor of 0.284 and t = 4.155. It shows that flow is a major factor that can enhance learner satisfaction. Thus, it can be seen that the three qualities and flow are important strategic means to increase learner satisfaction.

## 5. Discussion

### 5.1. Discussion

This study investigated the relationship between online remote lecture quality (system quality, information quality, and service quality), flow, and learner satisfaction for college students in Korea's COVID-19 pandemic situation. Online remote learning based on the LMS in universities is an unfamiliar challenge for professors and students. As it has been conducted for nearly a year, problems and areas to be improved have appeared, and at the same time, it is showing potential. As online remote learning of the LMS can be a means to supplement the limitations of face-to-face classes in the post-COVID-19 pandemic, its development and establishment will be necessary. The LMS has grown into an indispensable tool for facilitating teaching and learning in universities. Besides cultivating and improving university students' communication skills and assisting teaching in general, studies have also found that LMS implementation contributes to reducing the production of materials and preserving resources [63].

It was found that the system quality, information quality, and service quality of online remote learning quality positively affect flow. It was also identified that these three qualities have a positive effect on learner satisfaction. It was demonstrated that flow has a positive effect on learner satisfaction. This shows that flow and learner satisfaction are improved when the online distance learning quality is enhanced. In order to achieve the performance of non-face-to-face classes in the COVID-19 pandemic situation, it will be necessary to strengthen the quality of online remote classes of the LMS.

### 5.2. Implications

This study presented the implications based on the analysis results as follows. First, the quality of the remote lectures (system quality, information quality, and service quality) was found to positively affect flow. In particular, information quality was found to have the most influence on flow. This shows that university students value information quality more than system quality and service quality in the flow of online classes. In order to increase the flow of the students in online courses, class-related information, easy-to-understand content, and the latest information should be provided. Moreover, if universities improve remote classes' system quality and service quality, students' flow will also improve.

Second, the quality of online remote lectures (system quality, information quality, and service quality) was found to positively affect learner satisfaction. Information quality has been considered to play an important role in modern business, and the importance of information content is highly evaluated in e-commerce [35]. Information quality was also found to have the most influence on learner satisfaction. To increase the satisfaction of university students' online remote classes, the level of information quality related to classes should be raised, and system quality and service quality should also be improved from their point of view.

Third, flow was found to have a positive effect on learner satisfaction. This outcome is consistent with the results of previous studies that flow shortens the learning time, promotes active participation in learning activities, and positively affects learner satisfaction and academic achievement [53,54,57]. It is important to improve the overall quality to increase the students' flow of online remote lectures. To increase learner satisfaction, universities should provide sufficient and useful contents that learners need, and instructors should provide appropriate feedback related to interest and participation in the remote lectures. If an environment is created so that knowledge sharing among learners is actively carried out, the intensity of the learners' flow and satisfaction can be increased.

Universities' online remote learning caused by the COVID-19 is a landscape that is unfamiliar to professors and students, but rather, a high-quality learning method such as this can save time, money, and effort and can be seen as the basis for the development of a new learning environment. With the advancement of the fourth industrial revolution, online remote learning, which combines artificial intelligence and Internet of Things tech-

nology based on ICT, can be seen as an opportunity to be activated in universities and other educational institutions.

### *5.3. Limitation and Future Study*

The implications were presented based on the analysis results, but this study has limitations, so future studies will need to be supplemented in various aspects. First, in this study, data were collected only once, and the number of questionnaires used for empirical analysis were somewhat difficult to generalize. Second, research can be further advanced by developing various preceding and following constructs for online remote lectures. Finally, the study can be further expanded through hypothesis setting or additional analysis to confirm the difference between the remote lecture systems used or a real-time lecture and recorded lecture. Third, when evaluating the performance of online remote learning, if you compare and analyze practical subjects and nonpractice subjects, good implications can be obtained.

### 6. Conclusions

This study used the quality-flow-satisfaction model rather than the quality-satisfaction-flow research model to investigate how the quality of online remote learning affects flow and learner satisfaction in the COVID-19 pandemic. It was found that information quality has more influence on flow and learner satisfaction than system quality and service quality do. This shows that university students highly value information quality such as learning-related information, easy-to-understand educational content, and up-to-date information while participating in online remote learning.

In general, the formation of flow requires a long period. In this study, it was confirmed that information quality affects learner satisfaction more than flow does. This may be because the COVID-19 pandemic has lasted almost a year and a half, but the quality of online remote learning cannot be considered stable. If the flow of university students is properly formed in the online remote learning, the effect of the online class will be increased through self-directed participation.

There has been much discussion about online remote learning based on the LMS, and there have been various opinions about its performance. Many universities have steadily invested in online remote learning. Due to the COVID-19 pandemic situation, many universities have experienced trial and error while implementing online remote learning and are gradually improving the quality of the learning. In such an environment, the implications of the results of this study will be meaningful.

**Author Contributions:** J.-H.J. collected and analyzed the data and prepared the draft, and J.-I.S. organized the research design and finalized the paper. All authors have read and agreed to the published version of the manuscript.

**Funding:** This research received no external funding.

**Institutional Review Board Statement:** Not applicable.

**Informed Consent Statement:** Not applicable.

**Data Availability Statement:** Not applicable.

**Conflicts of Interest:** The authors declare no conflict of interest.

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
