# Peer review of "Assessment of University Students on Online Remote Learning during COVID-19 Pandemic in Korea: An Empirical Study"

_sustainability, doi:10.3390/su131910821_

Round 1

Reviewer 1 Report

What author mean by: " The types of online remote learning are learning management systems (LMS)"? This sentence is incomprehensible. I propose to develop it. 

Lack of information about: "In Korea, the online class method based on LMS was suddenly introduced" I propose to explain what the implementation of LMS was about. 

The paper lacks information on the characteristics of universities
 (private, public, size, type, location).

Lack of detailed characteristics of the students - respondents
(age, gender, does the student study extramural daily, do they work in the company)

Author did not characterize the questionnaire. What type of
survey was used in the research? At what time was the survey conducted and how the data was obtained?

Author states how many respondents took part in the survey (182 finally), but does not state how many universities students came from.

Author Response

Thank you for the opportunity to revise and resubmit our manuscript entitled “Assessment of University Students on Online Remote Learning during COVID-19 Pandemic in Korea: An Empirical Study.” Comments provided by each reviewer were very helpful in turning this research project into a stronger manuscript. Your comments and insights are greatly appreciated. Responses to the reviewer’s comments are outlined in the text boxes below. Changes made are represented by track changes in the manuscript.

Reviewer 2 Report

A very current topic that concerns studies around the world. Positively evaluates the performed research and its evaluation.
I have no further comments or recommendations on this article.

Author Response

Thank you for the opportunity to revise and resubmit our manuscript entitled “Assessment of University Students on Online Remote Learning during COVID-19 Pandemic in Korea: An Empirical Study.” Comments provided by each reviewer were very helpful in turning this research project into a stronger manuscript. Your comments and insights are greatly appreciated. Responses to the reviewer’s comments are outlined in the text boxes below. Changes made are represented by track changes in the manuscript.

Thank you very much for your positive evaluation of our paper. We will focus on correcting typos and editing to improve the quality of the paper.

Reviewer 3 Report

Dear Author(s),

There is extensive work done.

The literature review chapter (theoretical background) requires even more references to articles related to the topic.

It may be helpful if the methods (e.g. confirmatory factor analysis, structural equation model) used were described in more detail.

The manuscript, if the methods can be fully described, presents an interesting point for the research topic, providing a sound ground for future researchers in the field.

Congratulations on your interesting and insightful work. Looking forward to seeing its final form and also to see it published.

Kind regards,

The reviewer

Author Response

(The authors gave the same response as above.)
